# Multidrug-Resistant Biofilms (MDR): Main Mechanisms of Tolerance and Resistance in the Food Supply Chain

**DOI:** 10.3390/pathogens11121416

**Published:** 2022-11-24

**Authors:** Francisca A. E. de Brito, Ana P. P. de Freitas, Maristela S. Nascimento

**Affiliations:** Department of Food Engineering and Technology, School of Food Engineering, University of Campinas, Campinas 13083-862, Brazil

**Keywords:** foodborne pathogen, multidrug resistance, food safety, food hygiene, quorum sensing, recalcitrance

## Abstract

Biofilms are mono- or multispecies microbial communities enclosed in an extracellular matrix (EPS). They have high potential for dissemination and are difficult to remove. In addition, biofilms formed by multidrug-resistant strains (MDRs) are even more aggravated if we consider antimicrobial resistance (AMR) as an important public health issue. Quorum sensing (QS) and horizontal gene transfer (HGT) are mechanisms that significantly contribute to the recalcitrance (resistance and tolerance) of biofilms, making them more robust and resistant to conventional sanitation methods. These mechanisms coordinate different strategies involved in AMR, such as activation of a quiescent state of the cells, moderate increase in the expression of the efflux pump, decrease in the membrane potential, antimicrobial inactivation, and modification of the antimicrobial target and the architecture of the EPS matrix itself. There are few studies investigating the impact of the use of inhibitors on the mechanisms of recalcitrance and its impact on the microbiome. Therefore, more studies to elucidate the effect and applications of these methods in the food production chain and the possible combination with antimicrobials to establish new strategies to control MDR biofilms are needed.

## 1. Introduction

Antimicrobial resistance (AMR) is considered one of the 10 main threats to public health that humanity currently faces [1,2]. A microorganism that is resistant to at least one agent of three or more classes of antimicrobials is defined as multidrug-resistant (MDR) [3]. MDR pathogens cause infections refractory to antimicrobials, which limit the therapeutic options, and consequently result in worsening of the clinical condition of the patient, prolonged hospital stays with intensive care, and a high fatality rate [2,4]. The rapid spread of multidrug-resistant bacteria (MDR) has caused a global alert [4,5]. It is estimated that approximately 700,000 deaths from infections per year are caused by antibiotic-resistant strains, with an expected increase to 10 million in 2050 [6]. 

Given this scenario, a joint tripartite secretariat composed of the Food and Agriculture Organization of the United Nations (FAO), the World Organization for Animal Health (OIE), and the World Health Organization (WHO) was established to propose actions and goals for the control of AMR. Since 2015, this subject has been addressed in a more specific way, with the purpose of increasing awareness of antimicrobial resistance, and delaying the development and dissemination of resistant microorganisms.

For a long time, infections caused by MDR microorganisms were associated almost exclusively with hospital environments and nosocomial infections. However, it is now known that antimicrobial resistance is a problem that also encompasses the food production chain and can affect, in addition to risk groups, the general population [7,8]. Zoonotic MDR bacteria can be selected through the widespread or indiscriminate use of antibiotics in production animals. Vegetables and fresh products may also have contact with MDR microorganisms in the field through soil, water, or contaminated compounds. These microorganisms can also be transmitted by food handlers, facilities, equipment, and tools throughout the supply chain [9,10,11,12,13].

Several outbreaks involving food and MDR microorganisms have been reported in recent years, corroborating the need for greater attention being paid to these pathogens in the food production chain [14,15,16,17]. In 2014, an outbreak involving *Salmonella* Heidelberg in mechanically separated meat (MSM) from poultry was reported in the USA, in which 67% of the isolates identified were MDR [18,19]. In 2017, *Salmonella* isolates from an outbreak linked to papaya showed resistance to streptomycin and tetracycline [20]. In 2018, an outbreak caused by *S.* Reading in turkey meat resulted in 358 cases and one death; 64% of the isolates had antimicrobial resistance genes [21]. Furthermore, in Europe, high to extremely high levels of resistance to ciprofloxacin, nalidixic acid, and tetracycline were observed in isolates of *Campylobacter* sp. Outbreaks involving methicillin-resistant *Staphylococcus aureus* (MRSA) associated with chicken, turkey, beef, and/or pork meat have also been reported in Austria, Germany, the Netherlands, and Switzerland [22].

Another aggravating factor in the presence of these microorganisms in the food chain is the transmissibility of resistance genes in humans [23]. The transfer of antibiotic-resistance genes can occur in the intestine through the consumption of contaminated food [18]. Cabello [24] reported that the accumulation of resistant bacteria in the gastrointestinal tract may transform humans into a “bank of antibiotic resistance genes”. In addition, some studies have shown that zoonotic MDR bacteria can be easily transmitted to humans through food [25,26]. There is evidence of the transmission of mobile genetic determinants (plasmids, transposons, and cassettes of genes in integrons) in resistant pathogens from birds to humans [4].

Furthermore, some microorganisms have the ability to grow and form communities on biotic or abiotic surfaces [27,28]. These clusters of macromolecules and microorganisms are called biofilms. They are formed by bacteria, diatoms, fungi, and protozoa, and are immersed in an extracellular matrix (EPS) composed of polysaccharides, proteins, and exogenous DNA [29,30,31]. Biofilms confer advantages to cells as a survival strategy and adaptation to a variety of stresses. It is estimated that approximately 80% of bacterial infections are related to biofilms [32]. Biofilms resistant to antimicrobials are a significant issue for public health and food production chain. The recalcitrance of biofilms, a phenomenon that encompasses tolerance and resistance, confers to microbial communities the ability to survive and resistance to bactericidal agents [11,33]. Tolerance is more directly related to the characteristics of the EPS matrix and the physiological state of cells, whereas antimicrobial resistance refers to different mechanisms, such as extracellular signaling and horizontal gene transfer (HGT) [11,34,35,36]. Studies have suggested a possible correlation between antimicrobial resistance and biofilm formation capacity [37,38,39]. In addition, the patterns of resistance to multiple drugs presented by biofilms point to the need for a better understanding of the recalcitrance mechanisms [34]. Thus, the aim of this review is to present the main mechanisms of tolerance and resistance of MDR biofilms and their occurrence in the food production chain.

## 2. Biofilms and Antimicrobial Resistance

Biofilm formation comprises different stages (Figure 1). The first step is the reversible adhesion of cells to biotic or abiotic surfaces. This initial attraction occurs through Brownian motion, gravitational force, and chemotaxis mediated by nonspecific physicochemical interactions [40,41]. Anchorage may occur through adhesion of pili, flagella, fimbriae, or glycocalyx, and may be triggered by bacterial immobilization [42,43]. In this initial stage, only a small quantity of EPS is present [43,44]. The next step is irreversible adhesion, when there is an increase in the production of EPS, in addition to the formation of microcolonies [45]. Subsequently, cell multiplication and breakdown of enzymes and substrates begin. Thus, it acquires a three-dimensional and heterogeneous structure surrounded by pores and water channels, which are systems of entry and exchange of nutrients, oxygen, and metabolites, and a barrier against the penetration of sanitizers and antibiotics [31]; this stage is called maturation. The last stage is the dispersion stage, which can occur in two ways: passive or active dispersion [46]. In passive dispersion, large detachments of cells occur, which characterize the erosion of the biofilm. In active dispersion, isolated cells or small clusters of the biofilm are released [47,48].

The properties of the EPS matrix observed in biofilms, in addition to nutrient availability, synthesis and secretion of extracellular material, microbial interaction (competition, cooperation, or synergy), shear stress, and chemical (quorum sensing) or electrical (nanowire) communication are distinctive features of biofilms compared to free-living cells [31,49]. 

## 3. Mechanisms and Strategies for Recalcitrance

Recalcitrance is a term used to define the ability of pathogenic biofilms to survive, even in the presence of high concentrations of bactericidal agents. It comprises two independent phenomena: resistance and tolerance to antimicrobials as shown in Figure 2 [11]. In antimicrobial resistance, the cells not only survive but remain resistant, even after the biofilm is dispersed. In contrast, in tolerance, cells generally survive only while they are in the community, becoming susceptible to agents soon after dispersal [50,51].

### 3.1. Tolerance

Tolerance is related to environmental changes and external factors, and not necessarily to gene expression, as observed in AMR [34]. Such survival strategies have a temporal and conditioned character, and are mainly related to the characteristics observed in the EPS matrix and the physiological state of the biofilm cells. These mechanisms lead to the activation of a quiescent state, a moderate increase in the expression of the efflux pump, and a decrease in membrane potential [11,34,52].

The EPS matrix consists of different components, such as soluble gel-forming polysaccharides, extracellular proteins and DNA, amyloids, cellulose, fimbriae, flagella, and water. The adsorption sites in the EPS matrix act as adhesion points for exoenzymes, a factor that limits the transport of bactericidal agents and reduces their effectiveness by degrading metabolites [34,53]. In addition, the EPS matrix forms layers in the biofilm [54] that promote poor penetration of antimicrobials through the diffusion barrier, thus leading to entrapment, lower efficiency, or even inactivation of these agents [34,45]. The interference in the diffusion of the antimicrobials by the biofilms results in sublethal concentrations, which promotes microbial selection and the development of resistance. The state of cells, specifically the rates of slow growth and dormancy, is commonly related to the survival of biofilms exposed to antimicrobials [45]. As the biofilm matures, a greater number of cells reaches the stationary phase, making the biofilm less susceptible to these agents [55,56]. Cao et al. [57], showed that mature *P. aeruginosa* biofilms (7 days old) were up to 4 times more resistant to antibacterial action than young biofilms (1 day old). In addition, slow growth rates may lead to the establishment of a viable but non-cultivable state (VBNC) [58]. Another state, very common in resistant biofilms, is characterized by persistent cells, i.e., subpopulations of MDR bacteria within a biofilm that can differentiate into a protective phenotype [11,36,45,59]. In tolerance, changes in the biofilm microenvironment, such as in pH, osmotic stress, and substrate depletion, may be responsible for reducing the number of input channels and thus limiting the permeability of the cell membrane to an antibacterial, plus to reducing metabolic activity inducing the stationary phase [34,60]. In addition, the physical barrier promoted by the EPS matrix results in sublethal concentrations, and can promote microbial selection and emergence of sanitizer resistance in both MDR and non-MDR biofilms [61,62]. However, not all bacteria that show tolerance when in biofilm will show resistance after dispersion [11]. 

### 3.2. Antimicrobial Resistance 

Resistance to antimicrobial agents is the ability of some microorganisms to survive after being exposed to compounds considered lethal [45]. It is enabled by different mechanisms and can be classified into (I) intrinsic, (II) acquired, and (III) adaptive. Intrinsic resistance (I) encompasses all mechanisms of inhibition of agents that are inherent to the microorganism. In this type of resistance, strains may never have come into contact with the antimicrobial and still be able to survive. In acquired resistance (II), the process occurs through the incorporation of genetic material or mutations. This type is very common in biofilms, given the proximity of the cells in this environment [34,45,52]. Adaptive resistance (III) is the ability of a microorganism to survive due to changes in the expression of genes or proteins caused by exposure to environmental stress [63,64]. However, the latter can be a link between intrinsic and acquired resistance, since both genetic mutations and changes caused by environmental stresses can alter the expression of intrinsic defense mechanisms, and thus increase resistance to antimicrobials [11,34,36]. 

Quorum sensing (QS) and horizontal gene transfer (HGT) are the most commonly observed mechanisms in biofilms and may be involved in all three types of resistance. Nevertheless, other strategies to increase resistance to biocides can also be used, especially by MDR microorganisms, such as antimicrobial target modification, antimicrobial inactivation, and reduction in intracellular accumulation through regulation of efflux pumps and decreased membrane permeability [65,66]. In fact, these strategies are coordinated by QS or through HGT. 

#### 3.2.1. Quorum Sensing (QS)

Quorum sensing (QS) is a cell-to-cell communication system that mediates chemical signals allowing cooperative expression of genes. These genes are responsible for several physiological mechanisms, such as biofilm formation, sporulation, conjugation, motility, regulation of bacterial luminescence, and virulence factors, such as toxins, proteases, and adhesins [67,68]. QS is dependent on population density and is based on communication through autoinducers, signaling molecules that are recognized by cell surface receptors or in the cytoplasm. As they are produced, these molecules accumulate in the biofilm environment [7,69]. After recognition of the receptor, there is transcription involving genes that code for surface proteins, virulence factors, transcription factors, and proteins involved in biofilm development [70]. The autoinducers used by Gram-positive bacteria are peptides (AIPs), and those used by Gram-negative bacteria are acylated homoserin lactones (AHLs). For intra- and interspecies communication, autoinducer 2 (AI-2) is used in both Gram-positives and Gram-negatives [11,69]. QS positively influences recalcitrance (tolerance and AMR) [11]. However, most authors indicate a direct relationship and more significant action on AMR. Further, QS is an intrinsic resistance mechanism, but it also seems to exert an influence on adaptive resistance. Such thinking occurs because this mechanism allows transient adaptive changes in the expression of biofilm genes; however, this phenomenon can only occur through the previous existence of such genes [67]. Some studies found genes related to biofilm formation in MDR *S. aureus* (*icaA icaB icaD and icaC*), MDR *Salmonella* (*adrA, bapA, csgB, csgD, fimA, fimH*) and MDR *Listeria* (*flaA and luxS*) and observed strong biofilm production [13,71,72]. During exposure to stressors, QS promotes the overexpression of resistance genes, regulates the production of proteins involved in the permeability of the outer membrane, and increases the formation of micro-colonies, favoring biofilm formation and increasing sanitizer resistance [70,73]. Mutants with QS deficiency seem to produce less structured biofilms and are apparently more susceptible to antimicrobials since their structure is affected by this mechanism [11,34,45,72]. Furthermore, QS is directly linked to the regulation process of drug efflux pumps and modification of antimicrobial targets, important strategies of AMR [73,74,75]. These strategies have been previously associated with increased resistance to sanitizers and antibiotics [67]. In a recent study, the authors investigated the relationship between QS and MDR pathogens and suggested that biofilm formation by MDR bacteria provided greater antimicrobial resistance [76]. In addition, studies have tried to establish an association between AMR and increased biofilm formation [37,39]. The defense strategies coordinated by QS make the MDR biofilm a unique system with great resilience and a high potential for dissemination [77]. 

#### 3.2.2. Horizontal Gene Transfer (HGT)

Another mechanism widely recognized for adaptation in bacteria that is responsible for the increase and dissemination of resistance to antimicrobial agents by MDR biofilms is horizontal gene transfer (HGT). HGT occurs in acquired resistance through the incorporation of genetic material and the uptake of resistance genes or mutations. However, the genes responsible for coding AMR strategies are more expressed in gene transfer than in mutations [78]. Mobile genetic elements (MGEs) such as transposons, integrative-conjugative elements, integrons, or plasmids are easily disseminated from one bacterium to another through HGT [79], i.e., in biofilm HGT is responsible to spread the AMR among the community. Furthermore, HGT can contribute to MDR bacteria to acquire pathogenicity factors and adapt to different conditions [80]. The HGT mechanism consists of three main pathways: plasmid-mediated conjugation, transduction, and transformation; the first is most commonly observed in biofilms [81,82,83]. Plasmids are circular sequences of extrachromosomal DNA [78]. Plasmid-mediated conjugation is considered one of the main pathways for the dissemination of antibiotic resistance (ARG) genes. During conjugation, the transfer of resistance genes occurs laterally, by cell-to-cell contact, via pili or adhesin molecules.

In biofilms, whose populations are denser and more stable, there are higher rates of gene transfer. Factors such as high cell density and accumulation of mobile genetic elements contribute to the absorption of ARGs [84]. Plasmids can easily transfer ARGs between phyla and genera [67,85]. The T6SS secretion system, in which effector proteins and virulence factors are injected from the inside of a bacterial cell into another in a single step, provides an alternative mechanism of HGT and requires cell-to-cell contact. This fact emphasizes the proximity and the high cell density, common in biofilms, as important factors in bacterial resistance [86,87]. In fact, bacteria have macromolecular complexes capable of transporting various molecules from the cytosol to the extracellular environment. These secretion systems are classified into nine types: I, II, III, IV, V, VI, VII, VIII, and IX, also known as T1SS, T2SS, T3SS, T4SS T5SS, T6SS, T7SS, T8SS, and T9SS. Some of these systems differ in terms of the mode of operation and molecules that transpose [88,89,90,91]. Besides the T6SS the secretion, system type VIII (T8SS) has been related to biofilms. T8SS is composed of three protein components that facilitate the assembly of curli fibers in the extracellular environment and is used in the secretion of amyloid fibers by Enterobacteriaceae, which is one of the main components of biofilms [88]. This mechanism confers resistance to several antimicrobials [81,82,83,84,85,86,87,88,89,90,91,92,93]. Several studies have shown a strong relationship between antibiotic resistance genes and the ability of forming biofilm [94,95,96]. In a recent study, authors investigated a HGT encoded for antimicrobial resistance genes of MDR strains of *C. jejuni*, *E. coli*, and *S. enterica*. The results highlighted strong biofilm formation and promotion in HGT as being responsible for the evolution of the biofilm structure [82].

#### 3.2.3. Modification of Antimicrobial Targets

Some classes of antimicrobial agents use cellular components as targets. One of the strategies used by MDR bacteria is the modification of these components [97]. Modification of the target occurs through changes in the structure or the number of transpeptidases (PBPs) involved in the construction of the cell wall peptidoglycans. The increase in the number of PBPs leads to a decrease in the binding capacity of the antimicrobial and, consequently, to a reduction in the action of or resistance to this agent [77]. The acquisition of *mecA* genes in *S. aureus*, for example, promotes changes in the structure of PBPs and reduces the capacity of the antimicrobial agent [91]. The acquisition of *van* genes through HGT also confers resistance to vancomycin, as it results in changes in the structure of peptidoglycan precursors and reduces their binding capacity [1]. Among the most widely used antimicrobial agents in both animal and human therapy are *β*-lactams. Resistance to these agents may occur by interrupting the interaction between the target transpeptidases (PBPs) and the agent, among others [98]. Another widely used class of antimicrobials is fluoroquinolones, which act directly on the synthesis of nucleic acids. Bacteria modify the structure of DNA gyrase and topoisomerase IV to become resistant to fluoroquinolones [67,99]. Some antimicrobials also act by inhibiting binding to the active site of enzymes. The strategy consists of the expression of genes that promote modifications in these enzymes, interfering with antimicrobial binding but still allowing the binding of the natural substrate [67,97].

#### 3.2.4. Antimicrobial Inactivation

Another strategy adopted by MDR bacteria is the inactivation of antimicrobials, which occurs mainly in two ways: by transferring chemical groups to the antimicrobial or by complete degradation of the antimicrobial [97]. Inactivation by group transfer occurs most commonly through acetylation, phosphorylation, and adenylation, with the addition of acetyl, phosphoryl, and adenyl groups, respectively [63,100,101]. The synthesis of enzymes capable of hydrolyzing antimicrobials and thus inactivating them is coordinated by QS. These enzymes can be found intrinsically in the bacterial chromosome but can also be acquired through a plasmid [102]. Among the hydrolyzing enzymes, *β*-lactamases stand out [67,103]. When these enzymes confer resistance to next-generation cephalosporins, they are called extended-spectrum *β*-lactamases (ESBLs) and include members of the CTX-M, OXA, SHV, and TEM enzyme families [97]. Gram-negative bacteria such as *Pseudomonas* spp. and members of Enterobacteriaceae have *β*-lactamase chromosomal genes. Such genes are also found in Gram-positive bacteria such as *S. aureus*. Both groups have been widely associated with the production of biofilms [82,92,104].

#### 3.2.5. Reduction in Intracellular Accumulation

To avoid antimicrobial accumulation in the cell, bacteria reduce the agent content by decreasing membrane permeability and/or regulating efflux pumps. Porins control the selective access of small hydrophilic antimicrobial molecules to the periplasm by diffusion through a water-filled channel [105]. Mutations may alter permeability, but several genetic mechanisms may be involved in the synthesis of porins, reducing or inhibiting the process: premature stop codons, insertion elements, and negative regulation of expression [11]. The production of porins can act synergistically with efflux pumps, producing greater action against antimicrobials. Both have also been associated with biofilm formation [106,107].

Efflux pumps can transport a wide variety of compounds [11]. In this strategy, some proteins located in bacterial membranes function in the recognition and active transport of substances to the interior or exterior of the cell. The expression of genes encoding the pumpsis necessary and can occur in intrinsic resistance through the regulation of QS, in the acquired resistance through HGT, and in adaptive resistance when genes are induced or overexpressed after environmental stimuli or stress [11,63,97]. Efflux pumps play an important role in AMR and are divided into six families: principal facilitator (MF), extrusion of multiple drugs and toxins (MATE), small resistance to multiple drugs (SMDR), ATP binding cassette (ABC), efflux families of proteobacterial antimicrobial compounds (PACE), and resistance-nodulation-division (RND). The MF family consists of membrane proteins with 400 to 600 amino acid residues, and the MATE family has proteins with 400 to 700 amino acid residues. The SMDR family consists of smaller proteins with 100 to 120 amino acid residues organized as homodimers. Members of the PACE family can confer resistance to a range of biocides such as acriflavine, proflavine, benzalkonium, acriflavine, and chlorhexidine, and are encoded by many important Gram-negative human pathogens [11,108]. ABC transporters are active transporters formed by one or two proteins in addition to a dimeric cytoplasmic ATPase. Transporters of the RND family form a protein complex consisting of approximately 1000 amino acid residues [109,110].

In general, bacterial efflux pumps are composed of outer membrane channel proteins, fusion proteins, and cytoplasmic membrane efflux proteins. The fusion protein connects the outer membrane channel protein and the cytoplasmic membrane efflux. The expression of specific genes of the efflux pump can be regulated by QS, which has been demonstrated in some recent studies [111,112]. In addition, several studies have endorsed the relationship between efflux pump production, biofilm formation, and AMR [64,109,113]. According to Zhao et al. [102], some genes responsible for encoding efflux pumps in different transporter families are upregulated in biofilms. Baugh et al. [113], evaluating the inhibition of multidrug efflux as a strategy to prevent biofilm formation in *S.* Typhimurium, observed that the exposure of cells to efflux pump inhibitors reduced the formation of biofilms, which was also previously observed for *K. pneumoniae* and *P. aeruginosa* [106,114]. Efflux pumps are also related to resistance to sanitizers. Lister et al. [115] observed that MDR *P. aeruginosa* with RND family pumps was resistant to biocides. A recent study showed that several MDR pathogens, including *S. aureus, Proteus mirabilis*, and members of Enterobacteriaceae, have AmvA homologs (*QacA, SmvA*), which confer greater resistance to biocides (particularly chlorhexidine) [116]. Overall, these studies point to the direct contribution of efflux pumps in the AMR and its relationship with biofilm formation.

## 4. MDR Biofilms in the Food Production Chain

The food production chain is composed of several steps that go from farm to fork. The indiscriminate use of antibiotics in animal production, whether for therapy, prophylaxis, or growth promotion, has directly influenced food safety, since these pathogens can be transmitted from feces or manure of contaminated animals to farm workers, vegetables, water, or soil [9,11,47]. Jones-Dias et al. [117] reported the contamination of farm soil with resistant *E. coli* and the transfer of the mobile *bla* gene. Class 1 integrons have played a major role in the spread of antibiotic resistance. The presence of this integron has been reported in an *Enterobacter cloacae* strain isolated from baby spinach leaves. This element was previously described in human microbiota. The authors suggested that the integron migrated to anthropogenic environments through food [80]. MDR pathogens such as *Salmonella*, methicillin-resistant *S. aureus* (MRSA), *Listeria monocytogenes*, *Campylobacter jejuni*, *E. coli* O157∶H7, and Vancomycin-resistant enterococci (VRE) have been associated with foodborne outbreaks [15,16,118,119]. Furthermore, in the USA these MDR microorganisms are estimated to cause approximately two million illnesses and 23,000 deaths each year [120].

The pathogenicity and difficulty in treating diseases caused by MDR pathogens is not the only worrying factor about these microorganisms. The dissemination of MGEs has gained attention in the food supply chain [5,39,82]. MGEs such as plasmids, transposons, integrons, and gene cassettes—easily transferred in HGT—have potentiated the acquisition of pathogenicity factors by different bacteria and increased the adaptability of MDR foodborne pathogens to different niches of the food supply chain [80,117,121]. In addition, MGEs can encode resistance to various antibiotics from contaminated food to gut microbiota of patients infected with MDR bacteria [5]. Glenn et al. [122], demonstrated the presence of resistance genes that conferred phenotypic resistance against chloramphenicol (floR and cmlA), tetracycline (tetA, tetB, and tetG), sulfonamides (sull1 and sul2), ampicillin (temB), *β*-lactamase (bla PSE-1), and aminoglycosides (strA and aadA) in *Salmonella*. In addition, the antibiotic resistance genes (bla PSE-1, sul1, flor, aadA, and tetG) detected in the study were potentially transferable. The presence of MDR *Salmonella* strains in foods, such as poultry meat, is a major concern, considering their extremely resistant phenotypes and several transferable determinants (resistance genes to plasmid- and integron-mediated antibiotics) [39]. Bersot et al. [123] evaluated the antimicrobial resistance of *Salmonella* in the pork production chain in Southern Brazil and reported a high incidence of MDR strains. In addition, the authors pointed out the need for greater monitoring of antimicrobial resistance in the pork production chain and the possibility of these strains to be transmitted to humans. Furthermore, MDR enterococci are found extensively in food samples. Chotinantakul et al. [124] characterized the phenotypic virulence factors and the ability of horizontal gene transfer of a streptomycin resistance gene among enterococci isolated from fermented pork. The results demonstrated that the aadE gene can be transferred via conjugation between enterococci isolated from pork, contributing to streptomycin resistance. In addition, the authors highlighted the importance of HGT within the food chain and that transference to humans might be possible, causing harm or untreatable diseases. Indeed, the transmission network does not end at the consumer. The selection and proliferation of these MDR microorganisms can continue through environmental contamination and return to the primary production through effluents and human and animal fecal matter, resuming the transmissibility cycle [5,125].

MDR foodborne pathogens can spread through processing lines from raw materials, workers, and soil or water reaching end products. In 2019 in the USA, a MDR *Salmonella* outbreak linked to raw chicken products resulted in 129 cases and one death. *Salmonella* Infantis was isolated from live chickens, raw chicken products, and in slaughter and processing establishments [126]. From 2017 to 2019, a multi-country MDR *Salmonella* outbreak with 358 cases and one death was associated with raw turkey products. The pathogen was recovered in raw turkey products from 24 slaughter and 14 processing establishments [127]. Another relevant aspect of MDR foodborne pathogens is their ability to form biofilms. Biofilms are a major problem in the industrial environment of food production [31,47]. Formation and development of these structures have been reported in some types of food and food processing plants (Table 1), such as meat [12,123], poultry [13,128], dairy [129,130], seafood [131,132], vegetables, fruits, and their products [133,134]. These biofilms can be formed on diverse materials (stainless steel, PVC, polyethylene, polypropylene, wood, rubber, and glass), remaining viable for a long time in the food processing environment [31,135,136]. In fact, given the emergence of MDR bacteria to public health and the difficulty to remove biofilms from food manufactory plants, it is necessary to establish a possible relationship between AMR and biofilm formation in this sector. However, there are still few studies that identify virulence and resistance genes and associate them with the production of biofilms isolated from the food production chain, especially in the processing environment and facilities.

One of the first studies to assess whether strains resistant to antibiotics had the ability to form biofilms was performed in 2008 in a hospital setting. Kwon et al. [143] evaluated the relationship between methicillin resistance and biofilm formation by S. aureus strains. This study demonstrated a higher rate of biofilm forming in MDR strains compared to no-MDR. Tahaei et al. [104] evaluated the biofilm-forming ability of sensitive (MSSA) and methicillin-resistant (MRSA) biofilms. Although no direct association was found between methicillin resistance and biofilm formation, there was a positive relationship for biofilm formation for other antibiotics, such as erythromycin, clindamycin, and rifampicin. In the food production scenario, in Italy, in a study conducted in the pig production chain, 77.0% of the S. aureus isolates were MDR, and 55.2% were able to produce biofilms after 24 h [32]. Olowe et al. [141] isolated 216 *E. coli* strains from animals, humans, and food. Thirty-seven of the 216 isolates were MDR and showed the ability to form biofilms. It was also observed that all genes associated with biofilm formation were detected in antibiotic-resistant isolates. The authors also concluded that antibiotic resistance plays an important role in biofilm formation [141].

In China, 176 *P. mirabilis* isolates were investigated regarding the association between biofilm formation, virulence gene expression, and antibiotic resistance. The results indicated that biofilm formation was significantly associated with resistance to doxycycline, tetracycline, sulfamethoxazole, kanamycin, and cephalothin. Approximately 76.7% of the isolates were MDR or extensively drug-resistant (XDR). In addition, there was a high predominance of biofilm-producing isolates (92.05%), with a higher prevalence of virulence genes in the biofilm-producing group than in the nonproducing group [39]. In Poland, strains of MDR S. Enteritidis showed greater biofilm formation capacity [37].

In China, between 2014 and 2015, all *Salmonella* isolates from birds that showed biofilm formation capacity were resistant to ciprofloxacin, doxycycline, sulfamethoxazole-trimethoprim, ampicillin, and streptomycin, suggesting a relationship between biofilm formation and the MDR phenotype [144]. A study conducted in nine pig farms in the United Kingdom showed that most isolates of *S.* 4,[5],12:i:- (98.1%) and *S. Typhimurium* (76.7%) exhibited the RDAR phenotype (colonies red, dry, and rough), indicating the production of cellulose and curli fimbriae, which are related to biofilm formation. The presence of MDR strains capable of forming biofilms with enormous biomass production, such as *S.* 4,[5],12:i:- and *S. Typhimurium* DT104 reinforces the relationship between multidrug resistance and biofilm formation [145]. Ma et al. [82] investigated the HGT of chromosomally encoded antimicrobial resistance genes between *C. jejuni* mutants in biofilms. The authors highlighted HGT as a driving force for the spread of antimicrobial resistance genes among *C. jejuni* strains. It has also been suggested that the evolution in the structure of these biofilms may be directly involved in transmission to humans [146,147,148]. Several genes associated with biofilm formation, including genes involved in motility (*flaA*, *flaB*, *flaC*, *flaG*, *fliA*, *fliS*, *flgA*, and *flhA*), oxidative stress (trxA, trxB, ilvE, and nuoC), stress response (spot, csrA, ahpC, cosR, and cprS) and quorum detection (luxS) were also identified. In recent studies, Baaboua et al. [71] evaluated the biofilm formation for nine strains of MDR *C. jejuni* and *C. coli* isolated from food in Northern Morocco. The results demonstrated a strong relationship between the presence of antibiotic-resistance genes and the ability to build biofilm. Kaptchouang et al. [15] evaluated a total of 207 samples (meat, milk, vegetables, and water) and found that *L. monocytogenes* isolates were resistant to several antibiotics and showed a strong biofilm formation.

Although not fully understood, AMR seems to promote greater biofilm formation and consequently higher protection against the action of biocides commonly used in the food industry [149,150,151]. Wang et al. [138] evaluated biofilm formation, and antimicrobial and sanitizer resistance of MDR *S. enterica* isolated from beef plants. The results indicated that the strains were strong biofilm formers and highly resistant to quaternary ammonium chloride (QAC) and chlorine dioxide (ClO_2_), sanitizers widely used in the food industry. Obe et al. [13] isolated MDR Salmonella from poultry processing equipment after disinfection with chlorine and QAC. The authors observed strong biofilm-forming ability, and presence of specific genes for biofilm formation, sanitizer, and antibiotic resistance. Therefore, to reduce or minimize the occurrence of MDR foodborne outbreaks, the control and prudent use of antimicrobials in primary production, in addition to the implementation of Good Agricultural Practices (GAPs), are essential. In addition, cross-contamination must be avoided throughout the food supply chain through strict Good Hygiene Practices (GHPs), environmental monitoring, and control and prevention of biofilms.

## 5. Conclusions

Biofilm formation is a survival strategy influenced by several environmental factors that can enhance the survival capacity of various pathogens. However, these mechanisms are still poorly understood. In this review, we describe the different strategies encompassed in the mechanisms of resistance and tolerance to antimicrobials (activation of a quiescent state of cells, moderate increase in the expression of the efflux pump, decreased membrane potential, antimicrobial inactivation, modification of the antimicrobial target, and architecture of the EPS matrix). In addition, we demonstrate how some of these strategies have been related to biofilm formation and how they confer greater robustness to biofilms. Some lines of research aimed at interrupting the regulation of resistance genes, known as "Quorum Quenching", have been based on blocking the expression of target genes by the use of substances present in medicinal plants. However, there are few studies investigating the impact of the use of these inhibitors on recalcitrance mechanisms and their impact on the microbiome or even the use of strategies that combine other antimicrobials. In addition, more studies are needed to answer questions such as: How does the industrial environment influence the formation of these biofilms? Can the technological processes, common in the food industry, affect the expression of biofilm formation genes? Or what is their impact on the virulence and resistance of MDR pathogens? 

## Figures and Tables

**Figure 1 pathogens-11-01416-f001:**
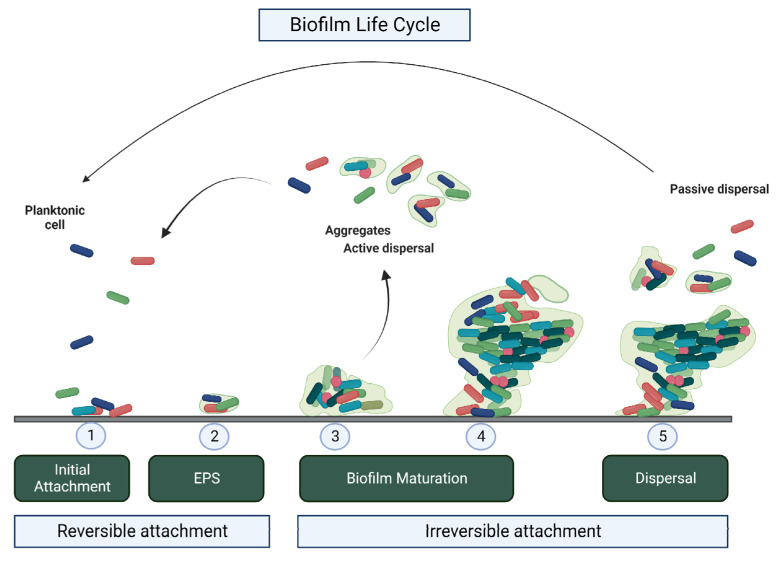
Microbial biofilm adhesion steps: 1—Start of bacterial adhesion (weaker forces); 2—Action of quorum sensing and beginning of stronger binding forces; 3—Formation of microcolonies with stronger EPS production and formation of three-dimensional structures; 4—Mature biofilm; 5—Detachment and reversion to planktonic growth (cell dispersion).

**Figure 2 pathogens-11-01416-f002:**
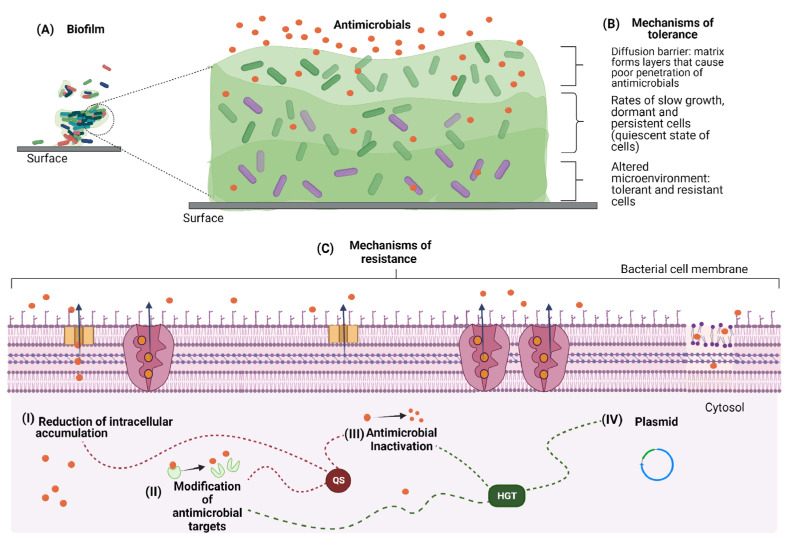
Mechanisms of recalcitrance. (**A**) Mature biofilm. (**B**) Mechanisms of tolerance: the extracellular matrix interferes in the permeation of the antimicrobial by the biofilm which cause microbial selection and development of resistance. These mechanisms lead to the activation of a quiescent state, a moderate increase in the expression of the efflux pump, and a decrease in membrane potential. Resistant bacterial cells (purple), sensitive bacterial cells (green). (**C**) Mechanisms of resistance. (I) Reduction in intracellular accumulation: bacteria reduce the antimicrobial content by decreasing membrane permeability and/or moderate increase in efflux pumps expression. (II) Modification of antimicrobial targets: the target alterations occur through changes in the structure or the number of transpeptidases involved in the construction of cell wall peptidoglycans. (III) Antibiotic degradation: resistance occurs by transferring chemical groups to the antimicrobial or by complete degradation of the antimicrobial. (IV) Plasmid: in biofilms, plasmid-mediated conjugation is considered one of the main pathways for the dissemination of antibiotic resistance (ARG) genes in HGT.

**Table 1 pathogens-11-01416-t001:** MDR bacteria reported in food and food processing plants.

Microorganism	Source	AMR Profile	Biofilm Formation	Biofilm Genes	Biocide Resistance	Reference
*Salmonella* sp.	Chicken slaughter plant	AMP; STX; CN; C; TET	Strong biofilm	-	-	[137]
*Salmonella* sp.	Beef slaughterhouse	TET; SFX; STR; CHL; AMP	Strong biofilms	-	Quaternary ammonium chloride; Chlorine and Chlorine dioxide	[138]
*Salmonella* sp.	Chicken slaughterhouse	AMC; AMP; FOX; XNL; CEF; CHL; CIP; FFN; GEN; NAL; NEO; STR; TET; SXT	Modest biofilm	-	Lactic acid and Cetylpyridinium chloride	[139]
*Salmonella*	Poultry processing equipment	STR; GM; TET; SF; CHL; GM	Modest biofilm and strong biofilm	*adrA, bapA, csgB, csgD, fimA, fimH*	Chlorine and quaternary ammonium compounds	[13]
*Listeria monocytogenes*	Chicken slaughterhouse	AMX; AMP; OX; CRO; VAN	Strong biofilms	-	*-*	[119]
*Listeria* sp.	Meat, milk, vegetables, and water	ERY; CM; CN; TET; NOV; OX; NA; K	Strong biofilm	*flaA* and *luxS*	*-*	[15]
*Staphylococcus aureus*	Milk	CRO; VAN; P	Strong biofilms	*icaA icaB icaD* and *icaC*	*-*	[72]
*Staphylococcus aureus*	Raw meat and meat products	P; AMP; TET; CN	Modest biofilm	*icaABCD*	*-*	[140]
*Campylobacter jejuni*	Chicken carcass	CEP; CEF; AMX; AGO; AZT; FLQ; NA; OXA; E; TET; STX.	Strong biofilm	*flaA, flaB, flaC, flaG, fliA, fliS, flgA,* and *flhA*	*-*	[71]
	Raw milk	CEP; CEF; AMC; AZT; FLQ; NAL; OXA; E; TET; STX.	Modest biofilm		*-*	
*Campylobacter coli*	Ground meat	K; STREP; TOB; CEP; CEF; AMX; AMC; AZT; STX.	Modest biofilm		*-*	
	Chicken meat	TOB; CEP; CEF; AMX; AMC; AZT; FLQ; NAL; OXA; E; STX.	Modest biofilm		*-*	
Vancomycin-resistant *enterococci* (VRE) and *Enterococcus* sp.	Poultry, pork, and meat products	TEC; C; ERY; Q/D; CP; AMP; TET	Weak biofilm or not biofilm formation	-	-	[17]
*Escherichia. coli*	Farm animals (goats, pigs, poultry, cattle, sheep) and animal products (milk, cheese, beef, chicken, yogurt).	TIC; TET; AMP; D; STX; LVX; ENR; CP; AMP; CXM; CEF; CZ; LEX; CAZ; TOB; CN; AMX; K	Modest biofilm	*agn43, bcsA, papC, csgA, fimH, fliC*	-	[141]
*Enterococcus faecalis* and *Enterococcus faecium*	Chicken and turkey processing plants	CHL; CP; ERY; FLA; GEN; K; L; NIT; P; Q/D; SM; STR; TET; TYL; VAN	Modest biofilm		-	[142]

AK: Amikacin; AMC: Amoxicillin-clavulanic acid; AMP: Ampicillin (ampicilina); AMX: Amoxicillin; AZT: Aztreonam; C: Chloramphenicol; CAZ: Ceftazidime; CB: Carbenicillin; CEF: Cefotaxime; CEP: Cephalothin; CFC: Cefaclor; CFM: Cefixime; CHL: Chloramphenicol; CM: Clindamycin; CN: Cefalosporins; CP: Ciprofloxacin; CRO: Ceftriaxone; CST: Colistin; CXM: Cefuroxime; CZ: Cefazolin; D: Doxycycline; ENR: Enrofloxacin; ERY: Erythromycin; F: Florphenicol; FFN: Florfenicol; FLA: Flavomycin; FLQ: Flumequine; FOS: Phosphomycin; FOX: Cefoxitin; FUR: Furazolidone; GEN: Gentamicin; GM: Aminoglycosides; K: Kanamycin; L: Lincomycin; LEX: Cefalexin; LVX: Levofloxacin; LZD: Linezolida; NA: Nalidixic acid; NEO: Neomycin; NIT: Nitrofurantoin; NOR: Norfloxacin; NOV: Novobiocin; OX: Oxacillin; OXA: Oxalinic acid; P: Penicillin G; Q/D: Quinupristin-dalfopristin; S: Streptomycin; SAM: Ampicillin-sulbactam; SF: Sulfonamide; SFX: Sulfisoxazole; SM: Salinomycin; STR: Streptomycin; STX: Trimethoprim/Sulfamethoxazole; TEC: Teicoplanin; TET: Tetracycline; TIC: Ticarcillin; TOB: Tobramycin; TS: Trimetroprim+Sulfamethox; TYL: Tylosin; VAN: Vancomycin; XNL: Ceftiofur.

## Data Availability

Not applicable.

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
