# Peer review of "Multidrug-Resistant Biofilms (MDR): Main Mechanisms of Tolerance and Resistance in the Food Supply Chain"

_pathogens, 2022, doi:10.3390/pathogens11121416_

Round 1
Reviewer 1 Report
The manuscript entitled “Multidrug-resistant biofilms (MDR): main mechanisms of tolerance and resistance and their impact on the food supply chain” have summarize the mechanism of biofilm formation and quorum sensing for MDR bacteria.
The study was written carefully and well in terms of language. However, this study requires minor corrections.
Minor revision:
In the Introduction, the Authors should briefly characterize the diseases caused by MDR bacteria, which is missing in.
Line 56: “Campylobacter” should be italic
Author Response
In the Introduction, the Authors should briefly characterize the diseases caused by MDR bacteria, which is missing in.
Thank you for your comment; a new sentence was added in the Introduction. Please, see lines: 30 to 32. “MDR pathogens cause infections refractory to antimicrobials, which limit the therapeutic options, and consequently result in worsening of the clinical condition of the patient, prolonged hospital stays with intensive care and high fatality rate [2,4].”
Line 56: “Campylobacter” should be italic ok
Sorry about this mistake, the change was made.

Reviewer 2 Report
The manuscript is well focused and organized. This is a topic of great interest, since antibiotic-resistant pathogens that are able to form biofilm are causing many diseases to become persistent and there may be a higher risk of death. So I just have a few suggestions to make to the authors.
1. Please use:
to format references in the text, especially when you use [...] at the beginning of the sentence. You should use Author name when you use citation at the beginning of a sentence.; examples to change: line 31, 59; 62; 334; 336; 342....etc.
2. Please replace "biofilm production" to "biofilm formation" throughout the manuscript
3. The title is "Multidrug-resistant biofilms (MDR): main mechanisms of tol-2 erance and resistance and their impact on the food supply chain" and is misleading because in fact the section about food chain is the shorter one. So, I suggest to change the title or give a deeper understanding of biofilm in food industry - not only cases about biofilm forming bacteria isolated from food supply.
Author Response
- Please use:
to format references in the text, especially when you use [...] at the beginning of the sentence. You should use Author name when you use citation at the beginning of a sentence.; examples to change: line 31, 59; 62; 334; 336; 342....etc.
We are very grateful for your comment. Changes in the references were made in accordance with the journal's guidelines in the entire manuscript. Please, see lines (67, 143, 340, 341, 427, 430, 436, 458, 469, 475, 479) .
- Please replace "biofilm production" to "biofilm formation" throughout the manuscript
Thank you for your comment; the change was made throughout the manuscript. Please, see lines (86, 92, 182, 203, 313, 351, 416, 428, 432, 440, 448, 450, 463, 471, 481).
- The title is "Multidrug-resistant biofilms (MDR): main mechanisms of tolerance and resistance and their impact on the food supply chain" and is misleading because in fact the section about food chain is the shorter one. So, I suggest to change the title or give a deeper understanding of biofilm in food industry - not only cases about biofilm forming bacteria isolated from food supply.
We are very grateful for your comment. We expanded the section on biofilm in the food chain, and also changed the manuscript title to “Multidrug-resistant biofilms (MDR): main mechanisms of tolerance and resistance, and some aspect on the food supply chain”. Please see section 4, lines 352 to 478.

Reviewer 3 Report
The manuscript reviewed available scientific data from various studies to establish a link between MDR bacteria and biofilm formation. Contrary to the title, the authors mostly discussed textbook information about the mechanism of antimicrobial resistance especially under section 4; antimicrobial resistance strategies.
Here the emphasis should be more on how the MRD bacteria help formation of biofilms. Section 5 is more relevant to the title of the manuscript. I would ask the authors to focus more on the information from studies where there is a data presented about biofilm formation by MDR bacteria or pathogens in diverse food supply chains such as meat, vegetable, or other foods processing plants.
Should expand more on section 5.
Line 46, there are multiple modes of transmission of bacteria within the food supply chain, importantly equipment used in the food processing plants. Need to include those mode of transmissions.
Combine subsection 3.2 with section 4 as it belongs there.
Author Response
Contrary to the title, the authors mostly discussed textbook information about the mechanism of antimicrobial resistance especially under section 4; antimicrobial resistance strategies. Here the emphasis should be more on how the MDR bacteria help formation of biofilms.
We are very grateful for your time and valuable suggestions. Section 3 was rewritten to emphasize MDR bacteria. In fact, in this section we intended to point out the two mechanisms that make up recalcitrance: tolerance and resistance, so we also changed the title of the section to “Mechanisms and strategies for recalcitrance”. Please see lines: 152 to 155, 177 to 178, 198 to 200, 206 to 214, 223 to 225, 232 to 253, 345 to 349.
Section 5 is more relevant to the title of the manuscript. I would ask the authors to focus more on the information from studies where there is a data presented about biofilm formation by MDR bacteria or pathogens in diverse food supply chains such as meat, vegetable, or other foods processing plants. Should expand more on section 5.
We appreciated your suggestion and contribution to our manuscript. Section 5 (now Section 4) was expanded. Although data on the subject is scarce, we added extra information on biofilm formation by MDR bacteria linked to food supply chain. Please, see Section 4, lines: 407 to 410. We also built a table with data on MDR biofilm forming bacteria reported in some types of food and food processing plants. Please see Table 1.
Line 46, there are multiple modes of transmission of bacteria within the food supply chain, importantly equipment used in the food processing plants. Need to include those mode of transmissions.
Thanks for your comment. Modes of transmission were better addressed in Section 4. Please, see lines 356 to 365, 369 to 419. We also complete information on the introduction, as follow: “These microorganisms can also be transmitted by food handlers, facilities, equipment and tools throughout the supply chain.” Lines: 50 to 51.
Combine subsection 3.2 with section 4 as it belongs there.
Thanks for your comment. Section 4 was merged into section 3.2 as suggested. Please, see lines 159 to 351.

Round 2
Reviewer 3 Report
the authors have addressed my comments appropriately, however, I would suggest the title should revised, remove the wording "some aspect",
instead replace it with the "in the food supply chain"
Author Response
Dear Reviewer 3,
Thank you for your comment. The title was changed as suggested.
Please, see the attachment.
Regards,
